# Evaluation of the Effects of a Bullying at Work Intervention for Middle Managers

**DOI:** 10.3390/ijerph17207566

**Published:** 2020-10-18

**Authors:** Elena Baixauli, Ángela Beleña, Amelia Díaz

**Affiliations:** Department of Personality, Assessment and Psychological Treatment, Faculty of Psychology, University of Valencia, 46010 Valencia, Spain; Elena.baixauli@uv.es (E.B.); Mangeles.belena@uv.es (Á.B.)

**Keywords:** middle managers, intervention, workplace bullying, psychosocial safety climate

## Abstract

The aim of the study is to evaluate the effects of a workplace bullying intervention based on the training of middle managers regarding bullying awareness, the consequences of bullying, strategies in conflict resolution and mediation/negotiation abilities. Overall, 142 randomly selected middle managers participated in the study. First, participants completed an information record and two scales assessing *bullying strategies*, *role conflict* and *role ambiguity*. The last two scales were completed again in a second phase three months after the intervention had finished. The intervention produced a decrease in the following bullying strategies: *effects on self-expression and communication*, *effects on personal reputation* and *effects on occupational situation and quality of life*, with all of the mentioned bullying strategies being suffered by part of the sample. In addition, the *conflict role* decreased in the group which received the intervention. Moreover, the decrease in the effects of the bullying strategy *effects on occupational situation and quality of life* was especially important in managers with higher responsibilities within the workplace. Results are discussed in the framework that (1) leadership practices and, more specifically, conflict resolution skills are strongly responsible for bullying at work; and (2) the importance of intervening in the early stages of the bullying process as a key element in the correction, but also as a potential prevention element, of bullying in the workplace.

## 1. Introduction

Bullying at work is a burgeoning research area in social psychology and more specifically in organisational psychology. Recent data about prevalence in Spain indicates that 15% of the workforce has suffered bullying in the workplace, mainly the fields of health and social care, education and state administration [1,2,3]. Data from other European countries estimates percentages from 4% in Portugal and Italy to 15 % in Finland with a mean of 9% [4]. The Trade Union Congress [5] reported that 29% of people have been the victims of workplace bullying in the United Kingdom, and the Public Service Union identified that one in four members across different public services [6], had experienced or had witnessed unfair discrimination in their workplace. Of these, 40% did not report the discrimination because of fear of being victimised or bullied further. The seriousness of the problem has prompted research on the subject in recent decades, with several studies clarifying the concept of bullying at the workplace [7,8,9,10]. These studies show a wide agreement presenting bullying at work as a situation in which one or more workers are repeatedly and regularly targeted with negative social acts over a period of time of at least six months [11]. Some characteristics should be present for a clear diagnosis: *frequency*, the situations or behaviours are pervasive and repeated quite often; *imbalance in power*, target and perpetrator are in different formal or non-formal power levels; and finally, the events are perceived by the victim as *badly intended* [12,13,14]. This situation leads to damaging effects on the victims and the organisation. The effects of bullying on the victims could vary from high levels of stress, anxiety and depression to Post-Traumatic Stress Disorder (PTSD) symptoms like intrusion and avoidance [15,16,17,18]. The organisations also experience the effects of bullying in terms of loss of productivity, absenteeism, sick leave, staff turnover, legal costs, and negative publicity [19]. The single factor of employees self-identifying as being bullied was associated with lower performance in the workplace with respect to those employees that did not self-identify as bullying victims [20].

Einarsen described three explicative models of bullying at work: the first one focused in the personality characteristics of victim and perpetrator, the second one had the specific organisational environment as the responsible of bullying, and the third one was based on the human relationships where conflicts and how to solve them are the protagonists [21]. Most of the research has been done in the second and the third models [15,22,23]. The second explicative model presented studies where the bullying at work was related to leadership style, role conflict, organisational climate, role ambiguity, challenging tasks, and work overload. Leymann found that inadequate leadership practices were common in bullying cases, and this finding led him to conclude that bullying is primarily a consequence of organisational conditions and that it cannot only be attributed to the intrinsic characteristic of individuals [11]. Leymann, thus lays the responsibility of bullying at work on the organisation and, more specifically, on the role of middle and top management. In a clear association, the third explicative model presented conflict as the central point in the human relationships. Conflict can always arise in the workplace, but it is the obligation of managers to oversee this situation according to the specific behavioural rules of the workplace. When the conflict is ignored, the bullying process deepens and worsens, often attributing the role of bullying precursor to the victim’s personality. Bowling and Beehr, through a meta-analysis, concluded that work stressors strongly contribute to workplace bullying [24]. More specifically, Hoel and Salin proposed that leadership practices are an important antecedent of bullying in the form of low satisfaction with leadership, autocratic leadership or “laissez-faire” leaders [22]. Similarly, two different studies concluded that the most consistently strong predictors of mistreatment at work occurs at the management level [25,26]. The intervention implemented in this study is based on the second and the third explicative models of bullying at work proposed by Einarsen [21] related to the management role played by middle managers, the way they manage conflicts, and the identification, and correction of bullying in workplace strategies.

In addition to leadership practices, role stress has been frequently related to two specific variables, role conflict and role ambiguity [23,27]. These variables are associated with stress and frustration and they increase the probability of triggering the search for scapegoats and result in bullying at work [28]. Role conflict happens when incongruent or incompatible tasks are required in the work role, such as receiving incompatible requests from two or more people. On the other hand, role ambiguity consists in the lack of clarity about the tasks required at work, such as to have unclear planned goals and objectives defined for the job. Merino and Forteza found high role conflict but low role ambiguity in middle managers [29]. These results were explained by the structural and functional character of the place middle managers have in the organisation. From a structural point of view, they are between the superiors and the subordinates, so the role they perform is clear; but from a functional point of view they have to transmit orders from the top management and have to organize the work of subordinates, so they are exposed to possible conflicts doing their work, hence the high role conflict. In a similar way, Hauge et al. using a big Norwegian sample, found that leadership practices and role conflict predicted bullying at work while role ambiguity did not when taking into account the effects of other predictors [30]. 

Bullying at work has also been described as an escalation process [13] where, before an employee is diagnosed as a bullying victim, he/she has gone through different stages. In the final phases of bullying, differences between bullying and interpersonal conflicts are very clear [14], but in the initial phases the differences are not so evident. The employee could be suffering different kinds of negative social behaviour, but these could initially not be intended as harmful, although their persistence underlines some kind of acceptance or even justification of the harm done to the victim [13]; the frequency could be low to medium, and the duration could be less than six months [11]. Furthermore, the imbalance of power as a basic element in the process of bullying could be informal and it depends on the ability of the victim to defend herself/himself from the bully [31]). This intermediate zone between the first conflicts, perceived as negative social behaviours by employees until the frequency of such acts begins to increase and becomes repeated and pervasive, should be the objective of an intervention, to be implemented as an efficient measure to stop an escalation of bullying in the future. Conversely, if the first complaint by those employees is ignored the situation will only get worse [32], highlighting the need to fill this gap in the training of managers at the intermediate levels. Therefore, the objective of the present work is to evaluate the effects of an intervention based in training middle managers in bullying at work strategies at this intermediate stage.

Taking into account the models that explain bullying at the workplace discussed above, the psychosocial safety climate theory [33,34,35] arose as “policies, practices, and procedures for worker psychological health and safety” ([34], p. 580). This theory works through three processes: *the mistreatment climate*, such as the identification of bullying strategies and proposing anti-bullying procedures; *work environment or design*, such as reducing stress factors affecting the work where role conflict and role ambiguity variables have a prominent place; and *conflict escalation*, presenting procedures to avoid conflict from escalating into bullying. The specific organisational climate proposed by the psychosocial climate theory is regulated by the management and leadership within organisations. Following the hierarchy in management, the role played by senior managers, middle managers and first line supervisors is crucial in the creation and maintenance of a safe and healthy climate at the workplace.

### 1.1. Middle Managers

Many of the studies cited above have been performed from the perspective of leadership management and it seems that middle managers play an important role in the developing of workplace bullying as either perpetrating, allowing, suffering or preventing the actions. However, from the perspective of bullying as an escalating conflict process [36], where the process could start with simple conflicts that could not be satisfactorily solved, managers again may play a crucial role. Salin proposed that potential bullies could be dissuaded from acting in organisations where managers are perceived to be prepared to intervene in bullying situations [8]. Conversely, victims of bullying reported lack of management intervention, attributed to managers’ poor training on the issue [37]. Going a step further, some authors [19,38,39] advised external consultants to carry out training in the organisation as a way to correct bullying at work. Fox and Stallworth also proposed alternative dispute resolution systems and workplace training as a way to handle bullying situations [40]. Key objectives included a clear definition of workplace bullying, the illegal character of the behaviour, consequences for the individual, organisation and society, role played by gender/race/organisational position in the bullying situation, possible responses by target and bystanders when the bullying behaviour happens and informal and formal solutions the organisation could propose. The aim was to provide specific guidance, skills and abilities to deal with bullying in internal (employees, supervisors, human resources departments, top managers and union representatives) and external (attorneys, physician and counsellors, psychologists and psychiatrics) environments. However, there is agreement that the key to success lies in the top manager’s hands [41], as expressed by Fox and Stallworth “on the extent that the top managers truly champion such initiatives” [40] (p. 238). In some countries, such as the United Kingdom, healthcare institutions propose early and informal interventions performed by managers to prevent disputes escalating into bullying [42,43,44]. In summary, both theory and research in workplace bullying present middle and top managers as a central element to break the development of bullying behaviours, but often fail to do so, in many cases due to a lack of specific training on the issue. The intervention we present here represents an attempt to fill this gap. 

### 1.2. Interventions in Bullying at Work

Bullying at work interventions could be classified according to the phase of the process they address, using the escalation conflict process as reference [36,45]. Accordingly, interventions in bullying at work would start with moderation in the first stages and finish with power intervention in the last ones. In addition, interventions can be classified in a multilevel model depending on the level where the bullying behaviours are happening. This model based in Heames and Harvey’s multilevel model of bullying [46], presents bullying intervention in the following levels: the victim/perpetrator in a dyadic level; the co-workers, managers, or fellow team members as witnesses or bystanders to the acts of bullying in a group level; and the organisation level when policy changes are necessary in the whole organisation [47]. In the dyadic level, mediation is proposed as an adequate intervention. For the group level, training and coaching are advised, and in the organisation level, new policy and an accompanying set of enforcement procedures should be implemented in order to obtain changes affecting all members of the organisation. An interesting aspect of this multilevel bullying intervention is that, if bullying at one level could affect the others, intervention at one level will also affect all other levels too. From a wider point of view, intervention can be classified as primary (preventative), secondary (ameliorative) and tertiary (reactive) [48]. Thus, interventions proposed in the first stages of bullying have a preventive/ameliorative character, so it is crucial to implement them before low-level conflicts develop into bullying, hence the importance of using them at the right time.

Gillen et al. propose four ways aby which intervention could work [49]: (1) reinforcing polices and culture of intolerance of bullying in the workplace involving employees, (2) using mediation and negotiation, when the first conflictive behaviours are identified to maintain a safe environment, (3) getting a risk assessment of job-related precursors of bullying at work, and (4) procuring awareness or education that incite the employees to rethink their interactive behaviour with colleagues. Our interest in this study is focused in the second and fourth ways. These include training middle managers in mediation, negotiation and conflict solving skills, giving them the ability to solve conflicts before they degenerate into bullying, making them aware of the consequences and legal repercussions of bullying behaviours, and above all, of their own important role as examples of civility behaviour. They should feel responsible for identifying and stopping incivility behaviours that may lead to bullying.

### 1.3. Bullying at Work Training Intervention 

The intervention proposed here is aimed at the first stages of conflict process [36,45], at the manager level [47], and at the second level (ameliorative) for smaller groups [48]. It is based on five anchor points. Firstly, it is based on Leymann [11], Einarsen [24] and Zapf and Gross [36] models of bullying at work, where the organisation and more specifically, the leadership practices and the conflict management are strongly responsible for bullying at work. It is also based on the different studies that place the spotlight on the middle managers, but not as perpetrators of bullying, but as victims [50] and key elements in the identification and correction of bullying in the early stages [8,41,43,44]. The third basis for the intervention is the multilevel model of Saam [47] which shows how changes in a specific level of the organisation will affect all other levels. The fourth basis is focused on the content proposed by Fox and Stallworth [40] for an intervention on workplace bullying designed to be applied in the organisation that could be very useful for middle managers. Finally, the fifth anchor point is the psychosocial safety climate theory [33,34,35], an integrating proposition that frames the bullying strategies as a *mistreatment climate*, highlights the role of variables such as role conflict and role ambiguity as *work environment or design*, and draws attention to the prevention of *conflict escalation* in order to avoid bullying. Moreover, these three processes are regulated by the hierarchy in management that is usual in most workplaces, from the senior manager, middle manager to the first line supervisors.

The intervention is targeted at a secondary level [51] when bullying strategies are at a low-to-medium level and it is usual to find social and organisational negative behaviours included in a list of bullying strategies [52], targeted at the victim frequently, although some isolated strategies could be of a higher level.

The intervention lasted 10 h/sessions distributed in two days and it was implemented individually (Table A1). The aim of the first session was to raise awareness of the workplace bullying issue. It involved explaining what bullying is, what kind of behaviours are included and aspects related to bullying such as duration, intentionality and negative consequences of bullying for people and business. An important element in this session is to make the participants aware of what constitutes bullying in the workplace and the illegal character of these behaviours. The second session was dedicated to conflict definition, features, stages, role conflict and conflict management. An important element at this point was to draw attention to the place that middle managers hold in the business, and how this position makes them the best to present model behaviours and to detect bullying behaviours. Middle managers could detect bullying behaviours in the workers under their supervision, in other middle managers, and the top direction, while sometimes suffering bullying behaviours from all of the above. Accordingly, their position is crucial in the detection of bullying in the workplace. The remaining eight sessions were dedicated to conflict resolution training and mediation/negotiation aspects including role-playing practice, emotion management in the workplace, communication and empathy strategies, and practicing mediator abilities of use in day to day situations. Participants were encouraged to use real conflicts they lived in their workplace as practical examples, or alternatively, the trainer introduced hypothetical examples in order to practice the skills and abilities to be learned. 

The aim was to train managers to solve conflictive situations with their subordinates, their equals or superiors using adequate strategies, and to teach them to identify strategies associated with bullying which should be avoided, corrected and never allowed in the organisation. Although the practice of mediation has been seriously criticized in bullying cases with the victim/perpetrator in a dyadic level [38,47,53,54], the training in mediator skills and abilities such as empathy, neutrality, respect for others, the ability to generate an environment of trust and to lower the aggressive tension between the participants in a conflict, or to allow the participants to define and clarify their positions, among others, have the potential to be extremely useful in the organisational roles of managers. 

In summary, taking into account the previous literature review in the field, the objective of this study was to evaluate the effects of the intervention descripted in Appendix A for middle managers, through the following four hypotheses:

**Hypothesis** **1** **(H1).**
*Taking into account that the study included randomly selected control and experimental non-clinical groups, we expect that only a portion of the whole sample will suffer bullying strategies, and this portion of middle managers will present a picture where some bullying strategies could be frequent and others are seldom or never presented.*


**Hypothesis** **2** **(H2).**
*All variables introduced in the study, role conflict, role ambiguity and bullying strategies, are related with bullying at the workplace, therefore, we expect positive and high relationships between all of them.*


**Hypothesis** **3** **(H3).**
*As the core point in the study, the implementation of a bullying at work intervention, we expect that middle manager suffering bulling strategies in the experimental group will show lower levels in the post-intervention phase, with no changes in the control group. Middle managers suffering higher levels of bullying strategies in the experimental group could benefit more from the intervention, showing the lowest levels of bullying strategies.*


**Hypothesis** **4** **(H4).**
*We expect that gender, institution/business size, public or private character and middle manager’s responsibility level in their work could affect the intervention results in the experimental group.*


## 2. Materials and Methods

### 2.1. Participants

The total sample was composed of 142 middle managers from different fields, including education and health (32%), private business (27%), state administration (21%) and NGOs (20%), all of them having given their consent to participate. With the agreement and authorization of the participating institutions, a schedule was established in which the intervention would be implemented. Experimental and control groups were also randomly distributed with 92 participants in the experimental group and 50 in the control group with the same number of men and women in each group. Due to the lack of relevant information in the scales by one woman in the control group, and the non-attendance to the two intervention days and the lack of important information in the scales by one of the men in the experimental group, the final sample included 140 middle managers. The experimental group was composed by 91 participants (46 women and 45 men) and the control group was made up of 49 participants (24 women and 25 men). The study was conducted in accordance with the Declaration of Helsinki, and the protocol was approved by the Ethics Committee of the University of Valencia (20160426).

### 2.2. Inclusion and Exclusion Criteria for Participation

The inclusion criterion was being a middle manager and the exclusion criterion was having problems speaking or understanding Spanish.

### 2.3. Instruments

Leymann Inventory of Psychological Terror, LIPT [52] validated in a Spanish population [55] with the original 45 bullying actions/strategies at the workplace made up the scale. The frequency of these actions/strategies has been measured using a Likert five-response scale (never, seldom, often, quite often, always). The five factors assessed were *Effects on self-expression and communication* (e.g., “You are yelled at and loudly scolded”); *Effects on social contacts* (e.g., “You cannot talk to anyone; access to others is denied”); *Effects on personal reputation* (e.g., “Unfounded rumours about you are circulated”); *Effects on occupational situation and quality of life* (e.g., “You are given tasks that affect your self-esteem”) and *Effects on physical health* (e.g., “Threats of physical violence are made”).

Role Conflict and Ambiguity in Complex Organisations Scale, RCA [56], adapted to a Spanish population [29,57], was used to assess *role conflict and role ambiguity*. It is comprised 13 items with a Likert seven-response scale (from 1—It does not represent my work at all, to 7—It represents perfectly my work). Eight items assess role conflict (e.g., “I have to do things that should be done differently”) and five items assess role ambiguity (e.g., “I know what my responsibilities are”).

The final information record concluded with the participants’ last four digits of their Identification Card, gender, and questions concerning their companies’/businesses’/institutions’ public or private nature, size, and responsibility level developed by the participants in their workplace.

### 2.4. Design and Procedure

An experimental design was used with researcher’s blind procedure, where the researchers have no information about the participants’ bullying strategies or conflict role and ambiguity role until the post-intervention data were obtained and introduced in the data file. This blind procedure preserves the research impartiality in the development of the intervention. 

When the permissions were obtained from the companies/businesses/institutions, the participants completed and signed the informed consent, the scales with clear instructions were given and the participants completed the scales for the first time. The intervention was carried out by the first author in the order that is presented in Appendix A. The scales were completed a second time by all participants three months after the end of the intervention. The participation in the study was anonymous, using the last four digits of the participants’ Identification Card as a way to match scales and intervention attendance. Statistical analyses were conducted using the IBM SPSS Statistics for Windows Software, Version 23 (IBM Corporation, Armonk, NY, USA).

## 3. Results 

There are two aspects related to the sample characteristics that should be commented on before the results of this study are presented. Both aspects are related to the LIPT Scale in the levels *often*, *quite often and always*, levels that represent people suffering bullying effects from a medium to a high level. More women than men are in the highest levels of this scale. Thus, more middle manager women are suffering more bullying than middle manager men. The second aspect concerns the bullying at work strategies reported. We found that basically, participants are suffering mainly from *effects on self-expression and communication* (N = 43) and *effects on occupational situation and quality of life* (N = 31). Only a smaller number of participants reported *effects on personal reputation* (N = 16) or *social contact* (N = 11), most of them reporting also effects in the previous bullying strategies cited. No participant reported *effects on physical health* in the response options of *often*, *quite often* and *always*. The individual bullying strategies more often reported were (1) “Your work is constantly criticized”, (2) “You are constantly interrupted”, (3) “Your superior restricts the opportunity for you to express yourself”, (4) “You are continually given new tasks”; (5) “Colleagues restrict your opportunity to express yourself”, (6) “You are yelled at and loudly scolded” and (7) “Your decisions are always questioned”. Five of them were from the factor *effects on self-expression and communication*. On the opposite side, we found some bullying strategies that do not represent the sample because no participant reported them more than *seldom*. This is the case of items representing the factors *effects on social contacts* and *effects on physical health*. From this information, we can say that the bullying strategies more often represented in the sample came from the factors *effects of self-expression and communication* and *effects in occupational situation and quality of life*. The *effects on social contact and physical health* do not represent our sample due to the very low frequency presented, and therefore, they have been analysed only as a part of the variable *total bullying strategies*. *Effects on personal reputation* has an intermediate place between the first and the last two factors, and thus is included in the following analyses. These are important aspects that have to be taken into account when discussing the results.

In the pre-intervention condition, there are no significant differences between control and experimental groups in the variables studied with a single exception, the variable *role ambiguity, χ*^2^ (22, *N* = 140) = 38.80, *p* < 0.05, is higher in the control group than in the experimental one. This difference remained in the post-intervention condition *χ*^2^ (22, *N* = 140) = 35.61, *p* < 0.05, showing that the variable has been hardly affected by the intervention. 

The reliability of the factors in each scale were analysed in the pre-intervention condition. The internal consistency of LIPT Scale’s factors was adequate for *effects on self-expression and communication* (α = 0.84), *effects on personal reputation* (α = 0.75) and *effects on occupational situation and quality of life* (α = 0.79), but reliability was low for *effects on social contacts* (α = 0.63). Due to the number of “never” answers for *effects on physical health* items, it was not possible to determine the reliability for this factor. RCA Scale’s factors showed also adequate reliability (α = 0.73 for *role ambiguity* and α = 0.84 for *role conflict*).

As for the participants who reported more than one bullying strategy in the levels *often*, *quite often* and/or *always* (LIPT medium-high subgroup), we found that 23 participants in the control group and 24 in the experimental group were in this situation. Those participants are the primary target of our analyses to show the effect of the intervention. Additionally, the highest point in the bullying strategies was analysed when percentile 75 was obtained, corresponding to 15 participants in the control group and to 16 in the experimental one (LIPT 75 Percentile subgroup). Accordingly, analyses are presented for the overall sample; for those people suffering from a medium to a high level of bullying at work strategies and, finally for those participants in the top level of bullying strategies, always in both control and experimental groups. There were more women than men suffering bullying strategies in these two situations, LIPT medium-high subgroup (30 women and 17 men) and LIPT 75 Percentile subgroup (22 women and 9 men).

Table 1 shows the Pearson correlations between the variables studied in the pre-intervention phase. All relations are positive and highly significant. The lowest value corresponding to the relationship between role conflict and role ambiguity (*r* = 0.26; *p* = 0.004). The relationships between role conflict with the remaining bullying strategies are higher than the corresponding to role ambiguity with bullying strategies. As it is expected, the different bullying strategies show high relationships between them.

Table 2 and Table 3 show results from the ANOVAs with repeated measures analyses for control and experimental groups in the three subsamples according to the bullying strategies level suffered. When the analysis is performed in the sample with all participants, including those with low or very low bullying strategies, there are no significant differences in the control and experimental groups between the pre-intervention and post-intervention phases for any of the variables assessed; however, looking at the means, it is noticeable that whereas in the control group most of the bullying strategies had higher levels in the post-intervention phase, they were lower in the experimental group, although not enough to show significant differences.

When only those participants suffering medium-to-high bullying strategies were included in the analysis, the picture was completely different. There are significant differences between the pre-intervention and the post-intervention phases in the experimental group in *effects on self-expression and communication* (*F*(1,23) = 5.39, *p* = 0.030; *η*^2^ = 0.19), *effects on occupational situation and quality of life* (*F*(1,23) = 6.58, *p* = 0.017; *η*^2^ = 0.22) and *total number of bullying strategies* (*F*(1,23) = 6.25, *p* = 0.020; *η*^2^ = 0.22). In these variables, participants get lower significant bullying strategies in the post-intervention than in the pre-intervention phase. The size effects of these differences are moderate. 

Concentrating on the highest point in the continuum of bullying strategies suffered, the percentile 75 subgroup shows significant differences between pre-intervention and post-intervention in the experimental group in almost all the variables analysed: *effects on self-expression and communication* (*F*(1,15) = 14.95, *p* = 0.002; *η^2^* = 0.52), *effects on personal reputation* (*F*(1,15) = 5.50, *p* = 0.034; *η^2^* = 0.28), *effects on occupational situation and quality of life* (*F*(1,15) = 10.17, *p* = 0.007; *η^2^* = 0.42), *total number of bullying strategies* (*F*(1,15 = 18.65, *p* = 0.001; *η^2^* = 0.57) and *role conflict* (*F*(1,15) = 6.46, *p* = 0.024; *η^2^* = 0.32). Again, the bullying strategies in the post-intervention are lower than in the pre-intervention phase. The size effects are large in *effects on self-expression and communication* and *total number of bullying strategies*, being moderate in *effects on occupational situation and quality of life*, *personal reputation and role conflict*. As it was expected, *role ambiguity* does not show significant differences.

A second ANOVA analysis included inter-group variables such as gender, business or institution size, business or institution public/private character and middle manager’s responsibility level. None of these independent variables produced significant interaction with the previous variables studied between the pre-intervention and post-intervention in the control or experimental groups, with the only exception of responsibility level developed in their work by the middle managers. A significant interaction was shown in the factor *effects on occupational situation and quality of life* in the experimental group with the manager’s responsibility level, in the comparison with all participants (*F*(1,89) = 4.28, *p* = 0.042; *η^2^* = 0.06) and also in the comparison with the participants suffering medium-to-high bullying strategies(*F*(1,22) = 4.72, *p* = 0.041; *η^2^* = 0.18). Whereas the variable *effects on occupational situation and quality of life* in managers with intermediate responsibility level showed similar scores between pre-intervention and post-intervention phases, those with high responsibility showed lower scores in *effects on occupational situation and quality of life* in the post-intervention than in the pre-intervention phase. Both interactions show a moderate size effect. The 75th percentile bullying strategies subsample was analysed using the non-parametric Wilcoxon signed-rank test, obtaining significant differences between pre-intervention and post-intervention in the experimental group with high responsibility for the factors *effects on self-expression and communication* (*p* = 0.028) *effects on personal reputation* (*p* = 0.026), *effects on occupational situation and quality of life* (*p* = 0.027) and *total number of bullying at work strategies* (*p* = 0.027), with lower effects in the post-intervention than in the pre-intervention phase. Figure 1 shows the effects on *occupational situation and quality of life* in the three comparisons: all participants, LIPT medium-to-high subgroup and 75th percentile subgroup in bullying strategies for medium responsibility and high responsibility in the experimental group between pre-intervention and post-intervention.

None of the cases, neither in the case of all participants, nor for participants who suffered medium-to-high levels of bullying strategies, nor in the case of participants from the 75th percentile, showed significant differences between the pre-intervention and post-intervention scores in the control group. In addition, none of the independent variables analysed produced any significant interaction in the control group. 

## 4. Discussion

The choice of middle managers as a sample in this study has determined that the effects of bullying strategies occurred mainly in *self-expression and communication* and *occupational situation and quality of life*. To them, the variable *role conflict* was added. The random selection of participants demonstrated that not all of them were the target of bullying activities. Because of these initial conditions the intervention was expected to have effects in middle manager suffering bulling strategies, mainly those suffering from medium-to-high levels. The results confirm that the intervention reduced the four factors significantly when pre-intervention and post-intervention three months after the end of the intervention were compared. The expected decrease was clearly noticeable, but only when participants who were suffering bullying strategies from a medium to a high level were analysed. These results confirm Leymann [11], Einarsen [24] and Zapf and Gross [36] models of bullying at work. Going further, studying a non-clinic sample suffering some of the bullying strategies from the LIPT scale, where few participants experience bullying in *personal reputation* and *social contact* and none of them suffered *effect on physical health*, the study shows that a key element in the fight against bullying is to intervene in the early stages of bullying at the workplace to avoid more serious consequences. Therefore, our results support hypothesis 1 and the relevance of the escalation process and the value to identify and correct the bullying strategies at the beginning, in the early phases of the bullying. [13,14,32].

Although all variables in the study presented positive and high relationships, confirming hypothesis 2, *role conflict and ambiguity* variables have shown a very different picture in the study. *Role conflict* has been affected by the intervention mainly in the participants who are in the top of bullying strategies, the 75th percentile. They were suffering more bullying strategies, but the intervention not only reduced these strategies but also *role conflict*. It seems that the training in middle managers has helped them to deal with incongruence or incompatibility in the tasks required of them. On the other hand, the intervention does not seem to have any effect on *role ambiguity*. These results confirm the importance of *role conflict* in middle managers, a variable they are more susceptible to suffer than *role ambiguity*. They are between the superiors and the subordinates, therefore, the role they perform is clear, hence its low level of role ambiguity; but from a functional point of view they have to transmit orders from the top management and have to organize the work of subordinates, so they are exposed to possible conflicts doing their work [23,29,30]. Therefore, hypothesis 3 has been confirmed only partially.

These results also support the important role of managers as important elements in the prevention, identification and correction of the early stages of bullying in the workplace [8,40,41,43,44,50]. Furthermore, taking into account the multilevel intervention model [47], where effects in a level will affect other levels, not only managers were suffering bullying strategies at a lower rate after the intervention, they would also be more aware and sensitive to those bullying strategies in their equals and subordinates and they will be more prepared to deal with conflicts and relations with their superiors. Likewise, the study supports the thesis of Fox and Stallworth [40] about the importance of an intervention on workplace bullying designed to be applied at different working roles in the organisation, where middle managers were included as key element.

Perhaps the more interesting aspect of our results refers to the role of manager’s responsibility level in his/her work. The intervention has been effective specially in the factor *effects on occupational situation and quality of life* for those who carry high responsibility in the workplace, that is, those that are middle managers are in a high organisation level. They are very close to top managers which have the power to introduce changes in the policy of the company, and a high possibility of success in correcting or even preventing the development of bullying in the organisation [40,41,43,44]. 

Although this research was not looking for gender differences, when the medium-to-high level or the 75th percentile subgroups were obtained in bullying strategies suffered, women were more represented than men in a significant proportion. While this result confirms previous findings in a Spanish population [58,59,60], it should be taken with caution due to the small number of participants. Gender differences in bullying at the workplace seems to be inconsistent and inconclusive with many elements affecting gender differences, such as sample’s nationality, sector where the participants are working, measure methods used in the study, sample’s social power, and the position in the organisation hierarchy responsibility level of middle managers is one of them [61]. Accordingly, hypothesis 4 has been confirmed only partially due to gender and middle manager’s responsibility level seemingly playing a role in the intervention implemented in this study.

It is noteworthy that the results presented in the study support the psychosocial safety climate theory [33,34,35] corroborating the importance of identifying and correcting bullying strategies to prevent the *mistreatment climate*, the important role played by role conflict in middle managers’ *work environment or design,* and the emphasis in the time when the intervention should be proposed in order to prevent *conflict escalating to* serious bullying strategies. Furthermore, a crucial point could be the implication of middle managers in the previous three regulating processes.

Managers have been under-examined in the bullying at work literature, being studied as perpetrators of bullying [62] but ignoring the role they could have suffering, preventing, detecting and correcting bullying behaviours in the organisation. Thus, engaging them in creating a respectful workplace through their training represents one of the best measures in eliminating bullying from organisations. Training middle managers on bullying characteristics, consequences and legislation, conflict resolution strategies, problem solving, supportive actions and mediation abilities provides them with the skills and abilities to manage stressors and bullying activities in their workplace [34].

Several limitations are present in this research. Firstly, the sample was selected from a Spanish population in the sectors of education, health, private business, state administration and NGOs, therefore, the results should be framed by these fields. Secondly, due to the fact that the sample suffered mainly from *self-expression and communication*, *personal reputation*, and *occupational situation and quality of life* bullying at the workplace strategies, the effect of the intervention in other bullying strategies has not been verified. Finally, the data have been obtained through self-reporting, which leads to possible biases. Future research should concentrate on interventions with longer tracing of participants in order to know if they have contributed to the establishment of long-term behavioural habits, making the workplace fairer and more respectful, and the organisation more efficient. 

The study contributes to the progress in the treatment of bullying at work, from a perspective of bullying as an escalation process with an exposure to systematic and frequent verbal, non-physical, and non-sexual aggressive behaviour at the workplace in the early stages [32]. The intervention we have implemented is addressed to the early stages of conflicts which have a reasonable probability to become severe bullying situations over time.

## 5. Conclusions

Our results show that, at the early stages in a bullying situation and mainly with *effects on self-expression and communication*, *personal reputation*, *occupational situation and quality of life* and *role conflict*, an intervention based on the training of middle managers, mainly those with higher responsibilities in their organisations, reduces bullying strategies and role conflict. This makes managers aware of bullying situations and more conscious about bullying consequences, inducing them to work as good role models and enabling them to detect and stop conflicts that could evolve into bullying situations. However, further work in the area would be appropriate for a deeper analysis. 

## Figures and Tables

**Figure 1 ijerph-17-07566-f001:**
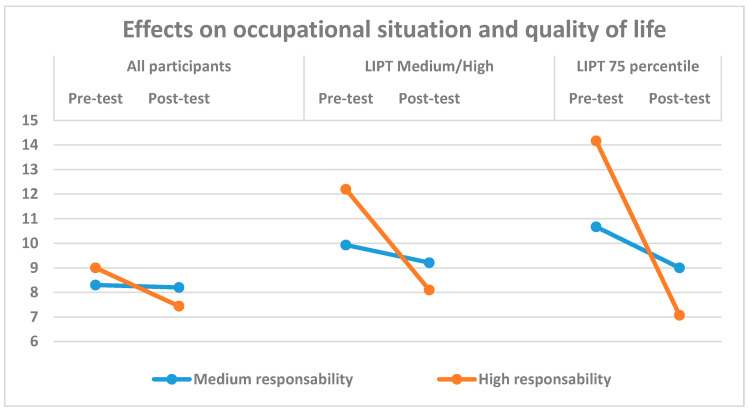
Intervention effects on occupational situation and quality of life in the experimental group.

**Table 1 ijerph-17-07566-t001:** Relationships between the variables.

Variables	1	2	3	4	5	6
1. Role conflict						
2. Role ambiguity	0.26 **					
3. Self-expression and communication	0.47 ***	0.39 ***				
4. Personal reputation	0.39 ***	0.29 **	0.62 ***			
5. Occupational situation and quality of life	0.42 ***	0.38 ***	0.65 ***	0.55 ***		
6. Total bullying strategies	0.49 ***	0.41 ***	0.92 ***	0.81 ***	0.83 ***	

** *p* ≤ 0.01; *** *p* ≤ 0.001.

**Table 2 ijerph-17-07566-t002:** Repeated measures analysis. Bullying strategies before and after the intervention for control and experimental group.

Group/Sub-Groups	Self-Expression and Communication							Personal Reputation						
	G	P	N	M	SD	F	Sig.	G	P	N	M	SD	F	Sig.
All	C	Pre	49	16.39	5.93			C	Pre	49	17.20	3.58		
participants		Post	49	16.59	6.41	0.27	--		Post	49	17.57	4.10	3.13	--
	E	Pre	91	14.55	4.54			E	Pre	91	15.87	1.61		
		Post	91	13.79	4.55	1.83	--		Post	91	15.64	1.91	0.66	--
LIPT	C	Pre	23	20.39	6.38			C	Pre	23	19.17	4.41		
Medium-		Post	23	20.09	7.65	0.20	--		Post	23	19.73	5.10	1.98	--
High	E	Pre	24	19.21	4.58			E	Pre	24	17.08	2.16		
		Post	24	16.29	5.87	5.39	0.030		Post	24	16.29	2.74	1.20	--
	C	Pre	15	23.13	6.33			C	Pre	15	20.93	4.51		
LIPT 75		Post	15	23.07	7.95	0.01	--		Post	15	21.80	5.21	2.03	--
percentile	E	Pre	16	21.33	21.33			E	Pre	16	18.20	18.20		
		Post	16	15.93	15.93	14.95	0.002		Post	16	16.07	16.07	5.50	0.034
**Group/Sub-Groups**	**Occupational Situation and Quality of Life**							**Total Bulling Strategies**						
	**G**	**P**	**N**	**M**	**SD**	**F**	**Sig.**	**G**	**P**	**N**	**M**	**SD**	**F**	**Sig.**
All participants	C	Pre	49	9.10	3.49			C	Pre	49	55.38	12.52		
		Post	49	9.06	3.42	0.27	--		Post	49	55.90	13.34	0.82	--
	E	Pre	91	8.58	2.86			E	Pre	91	51.21	8.22		
		Post	91	7.80	2.15	3.77	--		Post	91	49.63	8.62	2.19	--
LIPT	C	Pre	23	10.52	4.51			C	Pre	23	63.43	14.31		
Medium-		Post	23	10.44	4.43	2.10	--		Post	23	63.61	16.01	0.01	--
High	E	Pre	24	10.88	3.68			E	Pre	24	59.71	10.37		
		Post	24	8.75	3.02	6.58	0.017		Post	24	53.83	11.24	6.25	0.020
	C	Pre	15	12.33	4.67			C	Pre	15	70.73	12.72		
LIPT 75		Post	15	12.13	4.63	3.50	--		Post	15	71.07	15.15	0.05	--
percentile	E	Pre	16	12.07	4.06			E	Pre	16	64.40	7.15		
		Post	16	8.47	3.60	10.17	0.007		Post	16	53.00	13.29	18.65	0.001

G = group; P = phase; C= control group; E= experimental group; M = mean; SD = Standard Deviation; F = F ratio; Sig = Significance value; -- = non-significant.

**Table 3 ijerph-17-07566-t003:** Repeated measures analysis. Role conflict and role ambiguity variables before and after the intervention for control and experimental group.

Group/Sub-Groups	Role Conflict							Role Ambiguity						
	G	P	N	M	SD	F	Sig.	G	P	N	M	SD	F	Sig.
All participants	C	Pre	49	29.69	11.51			C	Pre	49	16.04	5.20		
		Post	49	29.39	11.81	0.35	--		Post	49	16.40	6.90	0.22	--
	E	Pre	91	29.24	10.50			E	Pre	91	13.09	4.41		
		Post	91	28.40	10.41	0.44	--		Post	91	13.66	4.35	0.01	--
LIPT	C	Pre	23	34.35	7.67			C	Pre	23	17.91	4.76		
Medium-		Post	23	31.58	9.90	1.20	--		Post	23	19.35	8.07	1.01	--
High	E	Pre	24	34.21	7.67			E	Pre	24	14.33	5.29		
		Post	24	31.58	9.90	1.29	--		Post	24	14.17	4.06	0.02	--
	C	Pre	15	38.27	7.77			C	Pre	15	19.33	4.48		
LIPT 75		Post	15	37.33	8.52	2.50	--		Post	15	19.93	4.62	2.39	--
percentile	E	Pre	16	35.87	8.63			E	Pre	16	14.53	5.64		
		Post	16	28.47	10.63	6.46	0.024		Post	16	14.20	4.36	0.02	--

G = group; P = phase; C= control group; E= experimental group; M = mean; SD = Standard Deviation; F = F ratio; Sig = Significance value; -- = non-significant.

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
