# Peer review of "Evaluation of the Effects of a Bullying at Work Intervention for Middle Managers"

_ijerph, 2020, doi:10.3390/ijerph17207566_

Round 1

Reviewer 1 Report

The issue of bullying features very strongly in the relatively new construct of PsychoSocial Safety Climate (PSC) wherein numerous studies have shown that bullying is one of the, if not THE most destructive element in the workplace environment. 

Further PSC studies show compellingly that PSC is a 'TOP DOWN' phenomenon and that change is never achieved until and unless there is a 'buy in' by the highest levels of management.

So it is surprising that none of this material has been acknowledged or integrated in this paper.

In my view this ommission means that the 'background' of the paper is seriously deficient. 

I would also have expected to seen Pearson's correlations reported to enable and ready overview of the interactions of variables.

These things being said, if the paper can be upgraded to place the study more completely in the fuller and most modern insights into the crucial issue of work place bullying it might be a useful addition to the literature on the area. 

Author Response

RESPONSES TO REVIEWER 1 (Responses written in bold)

Comments and Suggestions for Authors

The issue of bullying features very strongly in the relatively new construct of PsychoSocial Safety Climate (PSC) wherein numerous studies have shown that bullying is one of the, if not THE most destructive element in the workplace environment. 

Further PSC studies show compellingly that PSC is a 'TOP DOWN' phenomenon and that change is never achieved until and unless there is a 'buy in' by the highest levels of management.

So it is surprising that none of this material has been acknowledged or integrated in this paper.

In my view this omission means that the 'background' of the paper is seriously deficient. 

Authors thanks Reviewer 1 for his/her appropriate comments. The Psychosocial Safety Climate theory (PSC) has been included in the introduction, in the theoretical anchors of the study, and in the discussion sections and/or subsections. (Introduction section: page 3, lines 105-115; Introduction section, Bulling at Work Training Intervention subsection:  page 4, lines 184-189; Discussion section: page 11, lines 469-474). Psychosocial Safety Climate is now also in the list of keywords.

I would also have expected to see Pearson's correlations reported to enable and ready overview of the interactions of variables.

Pearson´s correlations between the variables have now been added to the manuscript with the inclusion of a specific hypothesis (hypothesis 2), Table 1 and specific comments in the Results and Discussion sections (Introduction section, page 5, lines 227-229; Table 1 in page 8; Results section, page 7, lines 335-340; Discussion section, page 10, lines 430-4319

These things being said, if the paper can be upgraded to place the study more completely in the fuller and most modern insights into the crucial issue of work place bullying it might be a useful addition to the literature on the area. 

Reviewer 2 Report

9_2020_ijerph-936649-peer-review-v1-1

Abstract

The purpose of reviewed paper is “to evaluate the effects of a workplace bullying intervention based on the training of middle managers regarding bullying awareness, the consequences of bullying, strategies in conflict resolution and mediation/negotiation abilities” (line numbers: 9-11). Authors describe research on negative social acts, the bullying at work – the effects of which are high levels of stress, anxiety and depression with Post-Traumatic Stress Disorder (PTSD) of workers. Organizations also have negative effects when their managers bullying at work – they are: decreased productivity and increased absenteeism, sick leave and staff turnover, as well as increased legal costs and a negative image in society. Bullying at work is organizational pathology affects from several to several dozen employees of the organization – depending on the country, where research on this social problem, was conducted. It is therefore an important and current management problem that needs to be addressed. Therefore, it is strength of work, because the best method of counteracting the workplace bullying is to increase the knowledge and awareness of managers and employees in this area. The area of research and conclusions as well as recommendations for managers fits very well with the subject of interest to the readers of the International Journal of Environmental Research and Public Health. Authors reviewed the literature, obtained the results of their own research and indicated recommendations for middle management managers – thus they showed the link between management theory and practice. The implementation of research results in organizations is important for achieving well-being of workers and fits in with the concept of corporate social responsibility.

Originality/Novelty

Authors explained the contribution of their work to the development of science as follows: “both theory and research in workplace bullying present middle and  top managers as a central element to break the development of bullying behaviours, but often fail to do so, in many cases for a lack of specific training in the issue. The intervention we present here represents an attempt to fill this gap” (line numbers: 128-131).

It has been known for a long time knowledge about the bullying at work and its negative effects on the employee, the employer and the whole organisation. However, the way in which this knowledge was presented in reviewed paper, was new and has a broad context and practical significance. It contributes to the elimination of organizational pathology, which is at the same time – a factor harmful to employees’ health and a social problem. It means, that the question is original and well defined. Therefore, results provide an advance in current knowledge.

Significance

The value and significance of the knowledge presented in the paper and Authors research experience are large.

Authors should complete the hypotheses. Besides, the work is methodically and substantively correct. All explanations are understandable and consistent.

Quality of Presentation

The structure of the literature review is correct and complete. In the theoretical part of the reviewed work, Authors presented the following issues:

  • the concept of bullying at work,
  • research and models of bullying at work,
  • contributing factors to bullying at work, e.g. personality characteristics of victim and perpetrator, ability of the victim to defend herself/himself from the bully, role conflict, role ambiguity, organizational climate, role ambiguity, challenging tasks, and work overload, leadership style, leadership practices, conflict management, the training of managers,
  • process of bullying at work,
  • symptoms of bullying at work,
  • effects of bullying at work,
  • role of top and middle managers in preventing workplace bullying,
  • interventions in bullying at work,
  • bullying at work training intervention.

Also the empirical part of the paper was written in an appropriate way, but Authors should complete the hypotheses. The data and analyses were presented appropriately. “The total sample was composed of 142 middle managers from different fields, including education and health (32%), private business (27%), state administration (21%) and NGOs (20%)” (line numbers: 208-209). The research was conducted in two groups: experimental and control, with the same number of men and women in each group (46 women and 45 men). The study was conducted in accordance with the Declaration of Helsinki, and the protocol was approved by the Ethics Committee of the University of Valencia.

The five factors assessed were effects on:

  • self-expression and communication,
  • social contacts,
  • personal reputation,
  • occupational situation and quality of life,
  • physical health.

An experimental design was used with researcher’s blind procedure. Research consent was obtained and that were anonymous. The main results and conclusions of the research are:

  • more middle manager women are suffering more bullying than middle manager men,
  • participants are suffering mainly from effects of the bullying at work on self-expression and communication and effects of the bullying at work on occupational situation and quality of life,
  • individual bullying strategies were identified that were more frequently reported,
  • the effects of bullying strategies occurred mainly in self-expression, communication, occupational situation and quality of life,
  • the intervention against bullying reduced significantly the four factors (three months after the end of the intervention were compared),
  • intervention in the early stages of bullying in the workplace and training middle managers are key in building a strategy against the bullying in work.

The ethical approach of Authors, during the research, should be highly appreciated.

Scientific Soundness

The study was correctly designed and technically sound. The analyses were performed with the highest technical standards.

It would be difficult, for another researcher, to recreate the research results, due to the original concept and implementation of the experiment. This should not be considered a weakness of the presented research. It’s related to the high level of complexity of this experiment, unique problems of the managers and other factors (e.g. cultural context).

The data was robust enough to draw the conclusions.

Interest to the Readers

Conclusions are interesting for the readership of the International Journal of Environmental Research and Public Health, in the opinion of the reviewer.

Overall Merit

The implementation of research results in organizations is important for achieving well-being of workers and fits in with the concept of corporate social responsibility. Work provides an advance towards the current knowledge and Authors have addressed an important long-standing question with smart experiments.

English Level

The English language is appropriate and understandable.

Author Response

RESPONSES TO REVIEWER 2 (Responses written in bold)

Comments and Suggestions for Authors

 Abstract

The purpose of reviewed paper is “to evaluate the effects of a workplace bullying intervention based on the training of middle managers regarding bullying awareness, the consequences of bullying, strategies in conflict resolution and mediation/negotiation abilities” (line numbers: 9-11). Authors describe research on negative social acts, the bullying at work – the effects of which are high levels of stress, anxiety and depression with Post-Traumatic Stress Disorder (PTSD) of workers. Organizations also have negative effects when their managers bullying at work – they are: decreased productivity and increased absenteeism, sick leave and staff turnover, as well as increased legal costs and a negative image in society. Bullying at work is organizational pathology affects from several to several dozen employees of the organization – depending on the country, where research on this social problem, was conducted. It is therefore an important and current management problem that needs to be addressed. Therefore, it is strength of work, because the best method of counteracting the workplace bullying is to increase the knowledge and awareness of managers and employees in this area. The area of research and conclusions as well as recommendations for managers fits very well with the subject of interest to the readers of the International Journal of Environmental Research and Public Health. Authors reviewed the literature, obtained the results of their own research and indicated recommendations for middle management managers – thus they showed the link between management theory and practice. The implementation of research results in organizations is important for achieving well-being of workers and fits in with the concept of corporate social responsibility.

Authors thank the rewarding words of Reviewer 2

 Originality/Novelty

Authors explained the contribution of their work to the development of science as follows: “both theory and research in workplace bullying present middle and top managers as a central element to break the development of bullying behaviours, but often fail to do so, in many cases for a lack of specific training in the issue. The intervention we present here represents an attempt to fill this gap” (line numbers: 128-131).

It has been known for a long time knowledge about the bullying at work and its negative effects on the employee, the employer and the whole organisation. However, the way in which this knowledge was presented in reviewed paper, was new and has a broad context and practical significance. It contributes to the elimination of organizational pathology, which is at the same time – a factor harmful to employees’ health and a social problem. It means, that the question is original and well defined. Therefore, results provide an advance in current knowledge.

Authors thank the rewarding words of Reviewer 2

Significance

The value and significance of the knowledge presented in the paper and Authors research experience are large.

Authors should complete the hypotheses. Besides, the work is methodically and substantively correct. All explanations are understandable and consistent.

Four hypotheses have been added to the manuscript following Reviewer 2 suggestion.

 Quality of Presentation

The structure of the literature review is correct and complete. In the theoretical part of the reviewed work, Authors presented the following issues:

  • the concept of bullying at work,
  • research and models of bullying at work,
  • contributing factors to bullying at work, e.g. personality characteristics of victim and perpetrator, ability of the victim to defend herself/himself from the bully, role conflict, role ambiguity, organizational climate, role ambiguity, challenging tasks, and work overload, leadership style, leadership practices, conflict management, the training of managers,
  • process of bullying at work,
  • symptoms of bullying at work,
  • effects of bullying at work,
  • role of top and middle managers in preventing workplace bullying,
  • interventions in bullying at work,
  • bullying at work training intervention.

Also the empirical part of the paper was written in an appropriate way, but Authors should complete the hypotheses.

As it has been stated previously, four hypotheses have been added to the manuscript following Reviewer 2 suggestion.

The data and analyses were presented appropriately. “The total sample was composed of 142 middle managers from different fields, including education and health (32%), private business (27%), state administration (21%) and NGOs (20%)” (line numbers: 208-209). The research was conducted in two groups: experimental and control, with the same number of men and women in each group (46 women and 45 men). The study was conducted in accordance with the Declaration of Helsinki, and the protocol was approved by the Ethics Committee of the University of Valencia.

The five factors assessed were effects on:

  • self-expression and communication,
  • social contacts,
  • personal reputation,
  • occupational situation and quality of life,
  • physical health.
  •  

An experimental design was used with researcher’s blind procedure. Research consent was obtained and that were anonymous. The main results and conclusions of the research are:

  • more middle manager women are suffering more bullying than middle manager men,
  • participants are suffering mainly from effects of the bullying at work on self-expression and communication and effects of the bullying at work on occupational situation and quality of life,
  • individual bullying strategies were identified that were more frequently reported,
  • the effects of bullying strategies occurred mainly in self-expression, communication, occupational situation and quality of life,
  • the intervention against bullying reduced significantly the four factors (three months after the end of the intervention were compared),
  • intervention in the early stages of bullying in the workplace and training middle managers are key in building a strategy against the bullying in work.

The ethical approach of Authors, during the research, should be highly appreciated.

Authors thank the rewarding words of Reviewer 2

 Scientific Soundness

The study was correctly designed and technically sound. The analyses were performed with the highest technical standards.

It would be difficult, for another researcher, to recreate the research results, due to the original concept and implementation of the experiment. This should not be considered a weakness of the presented research. It’s related to the high level of complexity of this experiment, unique problems of the managers and other factors (e.g. cultural context).

The data was robust enough to draw the conclusions.

Authors thank the rewarding words of Reviewer 2

 Interest to the Readers

Conclusions are interesting for the readership of the International Journal of Environmental Research and Public Health, in the opinion of the reviewer.

 Overall Merit

The implementation of research results in organizations is important for achieving well-being of workers and fits in with the concept of corporate social responsibility. Work provides an advance towards the current knowledge and Authors have addressed an important long-standing question with smart experiments.

 English Level: The English language is appropriate and understandable.

Authors thank again the rewarding words of Reviewer 2

Round 2

Reviewer 1 Report

It is pleasing to note that the authors have read the comments on their first draft carefully and responded appropriately. Given the mixed resuits it would be appropriate for them to include a suggestion in the Conclusion that further work in the area would be appropriate. 

Author Response

Following Reviewer 1 indication, the sentence:

“However, further work in the area would be appropriate for a deeper analysis” has been added at the end of the Conclusions Section